# Clearing Steatosis Prior to Liver Surgery for Colorectal Metastasis: A Narrative Review and Case Illustration

**DOI:** 10.3390/nu14245340

**Published:** 2022-12-15

**Authors:** Andrea Peloso, Matthieu Tihy, Beat Moeckli, Laura Rubbia-Brandt, Christian Toso

**Affiliations:** 1Division of Abdominal Surgery, Department of Surgery, Geneva University Hospitals, University of Geneva, 1205 Geneva, Switzerland; 2Department of Pathology and Immunology, University of Geneva, 1205 Geneva, Switzerland; 3Division of Clinical Pathology, Geneva University Hospital, 1205 Geneva, Switzerland

**Keywords:** protein diet, non-alcoholic fatty liver disease, liver surgery, nutrition, protein

## Abstract

Over recent years, non-alcoholic fatty liver disease (NAFLD) has become the most common liver disorder in the developed world, accounting for 20% to 46% of liver abnormalities. Steatosis is the hallmark of NAFLD and is recognized as an important risk factor for complication and death after general surgery, even more so after liver resection. Similarly, liver steatosis also impacts the safety of live liver donation and transplantation. We aim to review surgical outcomes after liver resection for colorectal metastases in patients with steatosis and discuss the most common pre-operative strategies to reduce steatosis. Finally, as illustration, we report the favorable effect of a low-caloric, hyper-protein diet during a two-stage liver resection for colorectal metastases in a patient with severe steatosis.

## 1. Introduction

In recent years, non-alcoholic fatty liver disease (NAFLD) has become one of the most common forms of liver disease in the Western world, accounting for 20 to 46% of liver abnormalities [1]. Additionally, NAFLD cases will likely continue to increase over the next 20 years, despite already affecting about a quarter of the world’s population. It is often associated with metabolic disorders, such as type 2 diabetes, hypertension, obesity and cardiovascular disease [2]. By being connected with these conditions, NAFLD, defined as the presence of >5% steatosis in the liver, can be identified as a hepatic expression of metabolic syndrome (MetS). In fact, approximately 90% of the patients with NAFLD have more than one feature of metabolic syndrome, and 33% have three or more criteria [3,4].

Nowadays, neoadjuvant or adjuvant treatments are routinely administered to patients undergoing surgery for colorectal cancer, including 5-fluoruracil, irinotecan and oxaliplatin. Chemotherapy-associated steatosis (CAS) is therefore increasingly frequent, potentially limiting surgical strategies [5,6]. Although surgical techniques and patient care have improved in recent years, hepatic steatosis is still recognized as an important risk factor for short- and long-term complications and death after general surgery, even more so after liver resection [7]. Similarly, steatosis also impacts the safety of living liver donation and transplantation [8]. Furthermore, it has been established that both metabolic syndrome and NAFLD stimulate the development of primary liver cancers [9,10] and also influence the metastatic potential of CRC [11].

Steatosis is reversible and has been the target of prehabilitation prior to surgery. Indeed, prior to bariatric surgery, a hypo-caloric, hyper-protein diet has become a standard practice to clear steatosis and promote liver shrinkage [12]. Living liver donors can also be managed preoperatively with a calorie-controlled diet, exercise or drugs to improve hepatic parenchymal quality [13].

The aim of this narrative review is to highlight the growing relationship between NAFLD and colorectal cancer. In particular, it assesses the literature on surgical outcomes after liver resection for colorectal metastases in patients with steatosis. It also aims to provide an overview of the most common pre-operative rehabilitation treatments targeting steatosis. Finally, as an illustration, we report on the favorable effects of a low-caloric, hyper-protein diet during a two-stage liver resection for colorectal metastases in a patient with severe steatosis.

## 2. Metabolic Syndrome and Non-Alcoholic Fatty Liver Disease

Metabolic syndrome describes a cluster of modifiable metabolic abnormalities that are associated with a globally increased risk of developing atherosclerotic cardiovascular disease, type 2 diabetes mellitus [14], neurological complications and cancer [15,16]. The definition of metabolic syndrome has undergone considerable changes, but the most recent consensus by the International Diabetes Federation in 2006 [17] defines metabolic syndrome by the presence of an increased waistline measuring >94 cm for men and >80 cm for women, along with the presence of two or more of the following: (1) blood triglycerides >150 mg/dL, (2) high-density lipoprotein (HDL) cholesterol <40 mg/dL in men or <50 mg/dL in women, (3) hypertension (>130/85 mmHg) and (4) elevated fasting glycaemia (>100 mg/dL (5.6 mmol/L) or diagnosed diabetes.

The metabolic syndrome epidemic represents a major challenge in the Western world, and it is estimated that 12–26% of the global population suffers from this condition [18]. NAFLD is considered as the hepatic manifestation of the metabolic syndrome, and it encompasses steatosis and progresses to liver fibrosis and, finally, to cirrhosis and end-stage liver disease [19]. The prevalence of NAFLD parallels that of the metabolic syndrome, and the rate of NAFLD is forecasted to double by 2030 [20].

Currently, the correct diagnosis of NAFLD is based on: (1) evidence of intra-hepatic fat accumulation (documented by imaging or histology), (2) the absence of significant alcohol consumption, (3) the absence of concomitant causes of hepatic steatosis and (4) the absence of co-existing causes of chronic liver disease [21]. Obesity is recognized as a risk factor for both steatosis and the development of colorectal liver metastases (CRLM). Given this, as well as the use of hepatotoxic chemotherapy regimens, steatosis can be found in up to 40% of postoperative surgical specimens derived from patients that have undergone liver resection for CRLM [22]. The histologic features of NAFLD and its developments include steatosis, ballooning, hepatocyte degeneration, inflammation, apoptotic bodies and Mallory–Denk bodies [23]. The level of steatosis severity varies considerably according to lifestyle, diet, duration and type of chemotherapy.

Steatosis can occur in two forms known as macrovesicular and microvesicular steatosis, which have distinct cytoarchitectural phenotypes. Macrovesicular steatosis displays a single large lipid droplet inside the hepatocyte with the nucleus displaced, whereas microvesicular steatosis is characterized by small lipid droplets with the hepatocyte nucleus located centrally. By definition, NAFLD steatosis is predominantly macrovesicular, with large droplets storing triglycerides [24], although microvesicular steatosis may also be present. HS is routinely classified in three tiers as mild (5 to 33%), moderate (33 to 66%) or severe (>66%) [23,25]. With mild steatosis, fat droplets appear to mainly have a zone 3 pericentral pattern, while more severe steatosis presents a panacinar distribution [26,27]. Steatosis is centered around the central vein, and periportal areas are classically preserved.

## 3. Molecular Pathways of NAFLD

Emerging evidence has spurred a considerable evolution of concepts relating to NAFLD pathogenesis and has called into question many previous notions. Historically, NAFLD lesions were considered to be the result of a double-hit condition [28]. The “first hit” involves fatty acids (FA) accumulation in the hepatocytes [29] increasing the vulnerability of the liver to many factors that constitute the “second hit” and promote hepatic injury, inflammation and fibrosis. Although the dictum was that steatosis (first-hit) always precedes inflammation (second-hit), different observations have challenged this view—for example, by showing that inflammation may precede FA intrahepatic accumulation and that steatosis may protect from liver damage, suggesting multiple hits acting simultaneously rather than sequentially to drive NAFLD progression [30]. In fact, beyond obesity, robust evidence suggests that a genetic heritability substrate, the gut microbiota and mitochondrial adaptation participate in NAFLD pathogenesis and progression.

### 3.1. Gene Variants

The genetic factors underlying the importance of inheritance in non-alcoholic fatty liver disease are starting to be understood [31]. Genetic variation, such as mutations or common polymorphisms, has been shown to be involved in the modulation of a range of risk factors, in part explaining the large spectrum of phenotypes. Four genetic variants have been most commonly described in association with NAFLD: patatin-like phospholipase domain-containing protein 3 (PNPAL3), transmembrane 6 superfamily member 2 (TM6SF2), glucokinase regulatory protein (GCKR) and membrane-bound O-acyltransferase domain-containing 7 (MBOAT7).

The magnitude of the genetic influence of PNPLA-3 on NAFLD outcomes is considerable and well documented. Due to its crucial role as a regulator in hepatocytes and stellate cells, PNPLA3 hydrolyzes triglycerides and catalyzes the transfer of the polyunsatured fatty acids (PUFA) from di- and tri-acyglycerols to phosphocholines [32]. The first association between this genetic variant (notably in its M-variants) was highlighted in 2008, when Romeo et al. [33] investigated 9229 gene variants in 2111 patients, whose liver fat content was determined by using proton magnetic resonance spectroscopy. They identified a single nucleotide polymorphism in the gene coding for PNPLA3, also known as adiponutrin. The exchange of an isoleucine for a methionine at position 148 (I148M) was strongly associated with the liver fat content in the different ethnic groups included in the study (overall *p*-value of 5.9 × 10^−10^). More recently, Dai G et al. highlight, through a meta-analysis including 14,266 NAFLD cases, how PNPLA3 was strongly linked with fatty liver and histological injury [34]. Additionally, G allele carriers were positively associated with hepatic steatosis (GG 3.24-fold and GC 2.14-fold versus the homozygous CC) independently of age, sex or BMI.

Among other genes, TM6SF2 is primarily implicated in VLDL secretion [35]. TM6SF2 is a regulator of liver fat metabolism influencing triglyceride secretion and hepatic lipid droplet content, and the loss of function due to a single nucleotide polymorphism (rs58542926) results in an increase in TAG levels and lipid droplet formation. For this, TM6SF2 inactivation has been positively correlated with the histological gravity of steatosis and all the elements of the NAFLD activity score [36,37]. NAFLD genetic susceptibility is gaining interest, even in normal-weight subjects, suggesting that NAFLD in lean individuals represents a distinct clinic-pathological entity [38]. Recent results show that lean people with histologically established NAFLD (known as lean NAFLD) had a larger proportion of the TM6SF2 5854296 T allele than overweight people with NAFLD, despite the distribution of the PNPLA3 rs738409 GG genotype being equal [39]. This demonstrates the importance of the genetic background, even in the absence of obesity. It should also be emphasized that this mutation appears to confer cardiovascular protection, thus questioning the relationship between NAFLD and cardiovascular disease. It is likely due not only to a downregulation of circulating apolipoprotein B but also to a complex anti-inflammatory effect [40].

GCKR is a key liver enzyme for glucose hepatic inflow. This modulates de novo lipogenesis by boosting the lipogenic pathway, providing a greater substrate for liver biosynthesis [41]. Several variants in the GCKR gene are reportedly associated with NAFLD. A genome-wide association analysis of a cohort including 19,381 individuals identified a strong correlation between GCKR (rs780094 variant) and a computed tomography-proven and biopsy-proven NAFLD (OR: 1.45, *p* = 2.59 × 10^−8^) [42].

Mainly expressed in the liver, MBOAT7 is a protein that is required for the phospholipid remodeling process, and its variant rs641738 predisposes to NAFLD [43]. Mancina et al. stratified the data by ethnic groups of the population-based Dallas Heart Study (DHS) and observed a positive significant effect (*p* = 0.019) of MBOAT7-rs641738 on hepatic triglyceride content that was restricted to African Americans. In contrast, the association with hepatic steatosis (NAFLD as a disease trait) remained significant in European Americans (OR: 1.37; 95% CI: 1.09–1.72; *p* = 0.007) [44].

### 3.2. Gut Microbiota and NAFLD

It is only during the past two decades that commensal bacteria that symbiotically live with the human host have received more attention, and this is rightfully so [45]. Especially, the gastrointestinal microbial community, known as the gut microbiota, plays an essential role in digestion, immunity and metabolism [46]. The liver is particularly exposed in this context, since it receives portal blood directly drained from the gastrointestinal tract and therefore is the first-pass organ for all intestinal bacterial products and metabolites.

In recent years, an altered gut microbiome has been increasingly associated with metabolic disorders, including NAFLD [47,48]. A strong relationship between NAFLD/NASH and the composition of the microbiome is suggested by several studies [49,50,51]. There also seems to be a correlation between the severity of NAFLD, the amount of fibrosis and decreasing heterogeneity of the gut microbiome [52,53]. The less diverse the gut microbiome is, the higher the risk for more severe NAFLD lesions and fibrosis. Additionally, specific genera such as Bacteroides and Ruminococcus have also been found to be related to steatohepatitis and fibrosis, respectively [49,52]. The metagenomics testing of stool samples from 56 obese individuals with steatosis revealed the enrichment of genes related to lipid metabolism, endotoxin biosynthesis and hepatic inflammation but was also evidence of aromatic disordering and branched-chain amino acid metabolism [54].

Overall, these studies suggest that the intestinal microbiota could contribute to NAFLD through several mechanisms:flattening of the microbial diversity [55];increasing levels of branched-chain and aromatic amino acids in the portal blood [51,56];increasing microbial production of metabolites such as phenylacetic acid and ethanol, which may increase liver lipid accumulation in vitro and in vivo [57,58];increasing in microbial endotoxins, which may contribute to inflammation [56];

Specific microbiota signatures may give us some further mechanistic insight into the interplay between the gut microbiome and pathophysiology of NAFLD. Specific signatures are observed for obesity [59], type 2 diabetes mellitus (T2DM) [60] and liver disease [61]. In the case of NAFLD, we observe a consistent altered signature including increased *Proteobacteria* [62] and *Enterobacteriaceae* [56], increased *Escherichia* and *Dorea* [63] and decreased *Coprococcus* [64] and *Provatella* [65]. The identification of specific signatures might represent a first step towards the development of a probiotic treatment preventing the progression of chronic liver disease. 

The number of clinical trials investigating the correlation between NAFLD and microbiota and even potential treatments is increasing [51,66,67]. However, it should be stressed that although the results are very promising, there are often large discrepancies between individuals. This can, in part, be linked to heterogeneous patient cohorts. In fact, sex, BMI, T2DM, patient populations (adult vs. pediatric), dietary intake or liver disease severity stage have a major impact on the microbiome. We therefore need more granular data on well-defined individual patient cohorts. This will represent a key next step towards a preventive probiotic supplementation.

That said, it must be pointed out that, at present, a bi-directional action between NAFLD and microbiota cannot be excluded. In fact, while bacterial dysbiosis likely contributes to a cascade of mechanisms finally leading to NAFLD, at the same time, NAFLD itself could induce bacterial dysbiosis. 

### 3.3. Hepatic Lipid Accumulation and Adipose Tissue Dysfunction

Lipids are the building blocks for cell membranes and essential for energy storage and cellular signaling. The liver serves as the master orchestrator of lipid metabolism and homeostasis [68]. It regulates the uptake, synthetization, storage and distribution of lipids in response to external signals such as fasting or feeding. These processes of lipid homeostasis are tightly controlled by hormones, transcription factors and nuclear receptors. A disruption of one of these pathways can lead to an imbalance between the hepatic uptake and clearance of lipids, which in turn can lead to lipid accumulation in the liver and eventually NAFLD. The hormonal environment that typically leads to such an imbalance of these pathways is characterized by hyperinsulinemia, hyperglucagonemia, growth hormone deficiency and hypercortisolemia [69].

In a metabolic state where glucose levels are high and fatty acids derived from dietary sources or the breakdown of triglycerides are not used for energy production, transcription factors sterol regulatory element binding protein 1C (SREBP1c) and carbohydrate response element binding protein (ChREBP) induce the expression of lipogenic genes in the liver. In the hepatocyte, fatty acids are esterified into cholesteryl esters by membrane-bound O-acyl transferases [70]. The neutral products of these acyl-transferases accumulate between the sheets of the endoplasmic reticulum until they eventually form a lipid droplet [68]. The mechanisms and regulation of the lipid droplet generation still hold many unsolved questions. A protein of the endoplasmic reticulum, fat storage–inducing transmembrane protein 2 (FIT2), is required for lipid droplet formation and might induce membrane asymmetries in the endoplasmic reticulum favoring droplet budding, but the protein is still poorly characterized [71]. Seipin, another protein, forms ring-shaped oligomers and stabilizes the connection between growing lipid droplets and the endoplasmic reticulum, thereby favoring the lipid droplet budding [72]. A knockdown of Seipin in mice leads to a significant decrease in adipose tissue mass and marked steatosis coupled with glucose intolerance and hyperinsulinemia [73].

Similar to fasting and exercise, hepatic lipid accumulation abnormally dysregulates adipose tissue lipolysis through insulin resistance and an increased sympathetic tone [74]. In obese individuals, the increased lipolytic profile coupled with an increased adipose mass leads to more adipose tissue-derived fatty acids entering the circulation [75]. On the other hand, Cortisol is chronically elevated in obese patients and plays an important role in the hormonal regulation of the adipose tissue [76]. Cortisol regulates lipogenesis and lipolysis differentially in adipose tissue based on the presence or absence of insulin [77]. Furthermore, an acute exposure to glucocorticoids does not induce lipolysis in adipose tissue; however, a chronic exposure to elevated glucocorticoids stimulates lipolysis in adipose tissue [78]. These factors lead to an increased delivery of adipose tissue-derived fatty acids to the liver of patients suffering from metabolic syndrome through the dysfunction of adipose tissue regulation and, eventually, an oversaturation of hepatocytes with fatty acids.

In summary, the hormonal dysregulation that characterizes the metabolic syndrome and obesity plays a key role in the increased accumulation of lipids in the liver. The presence of steatosis then reinforces the accumulation of hepatic lipids in a positive feedback loop.

### 3.4. Influence of Metabolic Disease and NAFLD on Colorectal Cancer

Elements of metabolic syndrome are significant risk factors for the development of colorectal cancer (obesity (BMI > 30) (OR:1.54) [79]; diabetes (OR:1.831) [80]). In addition, recent epidemiological studies highlight a relationship between NAFLD and the development of colon adenomatous polyps and the poor survival of patients with colorectal cancer [81].

Lee et al. investigated the colorectal neoplasm incidence in NAFLD patients through a population-based cohort study. In their analysis, after multivariate adjustment, NAFLD patients (n = 8,120,674) showed a significantly higher rate of colon cancer (HR:1.16). The data suggest that more active surveillance is needed for NAFLD patients compared to the non-NAFLD population [82]. According to Wu et al., NAFLD is also related to poor survival in patients with colorectal cancer [83]. They demonstrated that NAFLD patients with such cancer have a worse prognosis compared to controls, regardless of BMI or prognostic markers.

Interestingly, NAFLD patients also show a higher rate of colorectal liver metastasis (CRLM). Several clinical studies investigated the role of NAFLD as a potential driver of CRLM [Table 1]. Indeed, in parallel to the intrinsic mechanisms of metastasis (cellular dissemination from the primary tumor and awakening of dormant tumour cells), there is also bidirectional communication between tumour cells and the hepatic microenvironment [84]. Bauer et al. attempted to dissect how the liver microenvironment fatty changes impact hepatic metastasis using a choline-deficient high-fat diet with 0.1% methionine (CDAHFD) in mice. They showed that a moderate fatty liver has a protective effect against tumor growth, while more severe liver steatosis could stimulate tumor growth. In more detail, the authors suggested that CRLM proliferation is influenced by hepatic chronic inflammation (driven by increased levels of CD8+, INF-γ and TGF-β) and hepatic extracellular matrix remodeling [85].

Of interest, in 2019, Seki et al. investigated the role of the hepatic inflammasome (specifically, NOD-like receptor 4 (NLRC4) and IL-1) in colorectal cancer metastasis progression in a high-fat-diet mouse model. They reported that NLRC4 promotes tumor-associated macrophages polarization towards the M2 type, increases IL-1 and VEGF production and promotes colorectal cancer metastasis proliferation in the fatty liver [94]. Moreover, higher triglyceride levels, serum cholesterol and saturated fatty acids contribute to a pro-metastatic microenvironment via oxidative stress induced by reactive oxygen species (ROS) (including superoxide, hydroxyl radicals and H_2_O_2_) [95,96,97]. In summary, steatosis should be considered as an important risk factor for the development and progression of both primary and metastatic colorectal cancer.

HS and NAS scores have been also used to establish a prediction recurrence model after CRLM resection [98]. The authors included a total of 357 patients undergoing liver surgery for CRLM; among them, 42% presented no hepatic injury, 35% presented liver microsteatosis, 9% presented hepatic parenchymal disease, 4% presented steatohepatitis and 10% presented sinusoidal injury. Through the Fine–Gray regression model, severe hepatic parenchymal disease (NAS ≥ 3) was associated with an increased risk of intrahepatic recurrence (HR = 1.76, 95% CI = 1.07–2.90, *p* value = 0.027) and with a lower risk of extrahepatic recurrence (HR = 0.18, 95% CI = 0.04–0.75, *p* value = 0.019).

More recently, Yang et al. have investigated the impact of HS on CRC, especially on its stage IV, i.e., in the presence of metastatic disease, through a meta-analysis [99]. They finally analyzed data belonging to nine studies included (N = 14,197 patients). A pooled analysis of seven out of nine studies revealed that HS does not influence CRLM patient survival (HR: 0.92, 95% CI = 0.82–1.04, *I*^2^ = 82%, *p* value = 0.18), while it seems to improve cancer-specific mortality (two out of nine studies; HR: 0.85, 95% CI: 076–0.95, *I*^2^ = 41%, *p* value = 0.005). Additionally, HS is associated with a statistically significant reduction in disease-free survival in patients with CRLM (meta-analysis of four out of nine studies; HR: 1.32, 95% CI: 1.08–1,62, *I*^2^ = 67%, *p* value = 0.007).

Although these results are intriguing, they are contradictory, shedding light on the important knowledge gap as well as on the need for prospective studies to better assess the effect of HS on CRLM outcomes.

### 3.5. Chemotherapy-Associated Steatosis (CAS)

In current clinical practice, chemotherapy is often offered to patients suffering from colorectal cancer. Many involved hepato-toxic agents induce steatosis, steatohepatitis and sinusoidal injury and can impair liver function and regeneration [87,98]. IHF is the first sign derived from the hepatotoxicity of chemotherapeutic regimes.

Three different pathways have been confirmed for explaining the presence of steatosis: the excessive import of free fatty acids (FFAs), the diminished hepatic excretion of FFAs and impaired FFAs oxidation [99,100]. All of these mechanisms can be exacerbated by colorectal cancer-directed chemotherapy agents. Irinotecan, 5-fluorouracil (5-FU) and leucovirin are currently considered as the most common agents for colorectal cancer treatment [101]. Irinotecan is strongly associated with liver steatosis by affecting mitochondrial membranes and increasing toxic ROS species intermediates [102].

In 2017, Sommer et al. developed in vitro and in vivo models for 5-FU-induced steatohepatitis with the aim of identifying the underlying mechanisms for the induction of steatosis and its progression to inflammation. They reported that mitochondrial dysfunction is one of the major causes of steatosis and is driven by an increased expression of fatty acid acyl-CoA oxidase 1 (ACOX1), which catalyzes the initial step for peroxisomal β-oxidation [103]. Furthermore, 5-FU combined with leucovirin leads to steatosis [104], which significantly increases if irinotecan is added [105].

Chemotherapy duration impacts the manifestation of steatosis, and it is routinely considered that six cycles of chemotherapy are sufficient for its emergence [106]. To date, the benefits of chemotherapy clearly outweigh the risk of hepatotoxicity. Nevertheless, a comprehensive awareness of downstream complications is fundamental to the global management of these patients.

### 3.6. Impact of Hepatic Steatosis in Liver Surgery

Besides jeopardizing patients’ oncological outcomes for primary and metastatic colorectal cancer, there are growing data confirming the major role of hepatic steatosis in surgical outcomes. Additionally, for liver surgery, steatosis is a serious precondition in terms of perioperative outcomes [107,108,109] and has been comprehensively explored, as illustrated in Table 1. Berhns et al. [86] reported as early as 1997 that patients with steatosis (n = 135) had longer surgery times, higher rates of blood transfusion and higher post-operative bilirubin and AST levels. At the same time, larger studies established that HS is associated with higher rates of wound, hepatobiliary and gastro-intestinal complications in cases of hepatic resection for colorectal cancer metastasis [87].

Later, Kooby et al. [88] published the results of a retrospective matched analysis comparing the surgical outcomes of patients with mild (n = 122), moderate (n = 60) and severe HF (n = 12). All steatosis forms were associated with higher rates of hepatobiliary complications (such as cholangitis and ascites of hepatic abscess). In 2018, Sultana et al. recognized that, when associated with other metabolic syndrome components, steatosis increases the risk of liver failure after hepatectomy [110]. More recently, Fagenson et al. reported similar findings [89]. Their retrospective propensity-score-matched analysis investigated 2927 patients with steatosis and normal livers undergoing major hepatectomy (≥3 segments). The data not only confirmed steatosis as a risk factor for biliary complications but also highlighted its role in pulmonary complications.

Thus, to address the risk of morbidity and mortality in patients with steatosis, de Meijer et al. conducted a meta-analysis grouping 1000 patients from six different observational studies [109]. Compared to normal liver parenchyma, steatosis < 30% led to a significantly increased risk of postoperative complication, with an RR of 1.53 (1.27–1.85). Starting from 30% steatosis, the RR increased to 2.01 (1.66–2.44). Additionally, an HS ≥ 30% was associated with a higher mortality, with an RR of 2.79 (1.19–6.51).

In light of the above, steatosis of all forms of severity is an important factor in patient outcomes after surgery. Over the last 20 years, tremendous medical and technological progress has enabled increasingly aggressive liver surgery. However, these strategies remain strongly linked to the quality of the remaining liver parenchyma, which may lead to surgical planning to deviate from the standard guidelines. Indeed, steatosis accompanied by impaired lipid metabolism hinders liver regeneration ability [90,111,112]. This can be explained in part by the detrimental effects of steatosis on liver microcirculation and resistance to ischemic damage after significant parenchymal resection.

### 3.7. Clearing Steatosis Prior to Liver Surgery

Based on the studies mentioned, metabolic syndrome and steatosis negatively impact the short- and long-term outcomes after liver resection. Because the number of patients with steatosis is likely to increase in the coming years, it is of paramount importance to define measures to improve outcomes. While published evidence suggests that a 4-to-6-week interval is enough to clear part of the chemotherapy-induced liver toxicity [107], several studies have aimed to demonstrate the utility of other types of intervention to reduce steatosis content [113,114,115].

Steatosis is also considered one of the major causes of donor exclusion in living-donor liver transplantation programs [116]. Thus, several protocols have been proposed to reverse steatosis to enlarge the pool of liver living donors. To reverse steatosis and thus make 16 patients eligible for donation, Choudhray et al. suggested 1200 kcal/day and at least 60 min/day of moderate cardio training for at least 18 days [117]. This diet led to significant weight lost (7 ± 4.3 kg) in 15/16 patients, while 14/16 underwent living donation according to a second biopsy confirming hepatic steatosis reversal. Moreover, a complete normalization of liver parenchyma was observed in 7/16 candidates.

The decrease in steatosis following a preoperative diet has gained attention for being able to reduce bleeding during liver surgery. In a landmark study, Reeves et al. (2013) [91] reported that a one-week hypocaloric diet (900 kcal/day; 20–40% fat and 30–50% carbohydrate) reduced steatosis compared to control patients (15.7% versus 25.5%, *p*-value = 0.05). Fifty-one patients (BMI ≥ 25 kg/m^2^) (vs. sixty control patients) were assigned to a 900 kcal/day diet (composed of 33 g of protein, 18 g of fat for around 20–40% of total daily calories and carbohydrate for around 30–50% of total daily calories) or a normo-caloric diet one week prior to liver surgery. In detail, the diet plan consisted of three meals during the day: breakfast, lunch and dinner. Breakfast consisted of: oatmeal (1 cup) + milk or Slim-Fast^®^ (low carbohydrate; 12 oz) + fruit (1/2 cup); lunch consisted of: cottage cheese (3/4 cup) or Slim-Fast^®^ (low carbohydrate; 12 oz) + fruit (1/2 cup); dinner consisted of: lean meat or fish (3 oz) or Slim-fast^®^ (low carbohydrate; 12 oz) + fruit (1/2 cup). Dietary compliance was not assessed.

Following these findings, the same group explored the impact of a low-fat diet on intraoperative blood loss and outcomes after liver resection in a bi-institutional, surgeon-blinded, randomized prospective trial [92]. A total of 60 patients (BMI ≥ 25 kg/m^2^) were randomly assigned to an 800 kcal/day diet (20 g fat, 70 g protein) or normo-caloric diet one week prior to liver surgery. In detail, the low-calory diet plan provided five units of Optifast^®^ and an unlimited free-calories fluid intake one week before surgery. In the diet group, intraoperative blood loss was reduced (452 versus 863 mL; *p*-value > 0.005), and the liver was judged as easier to manipulate (following the Likert scale). Interestingly, no difference was detected in the level of steatosis, although there was a significant reduction in glycogen content in the liver parenchyma (PAS stain score 1.61 versus 2.46; *p*-value < 0.0001).

Besides life-style interventions, a plethora of pharmaceutical molecules have been tested to decrease steatosis, including liraglutide [118], pioglitazone [119] and ω-3 fatty acids [120]. One of the main drawbacks of the drug-based management of steatosis is the longer window needed to obtain significant results. Depending on the molecule used, this period ranges from 4 months to 1 year. Therefore, it is difficult to apply them in a short pre-operative timeframe. Combined, these data identify lifestyle interventions and dietary modification as important tools for decreasing steatosis in the available window of preoperative time.

Dietary interventions often consist of high-protein diets [121,122,123]. Amino acids, often referred to as the building blocks of proteins, are biochemical compounds addressing several crucial roles, including synthesizing proteins in cells [124]. It has recently been pointed out that proteins (and thus also the amino acids that make them up) also play an essential role in controlling the immune response [125]. This area, known as immunometabolism and intended as the modulation of immune response by nutrients, is gaining particular attention in the context of cancer therapies and immune-therapies.

Five aminoacids have been identified as potential important players in modulating T-cell proliferation and activation:the Cysteine-Cystine-Glutathione Axis [126,127,128,129]L-taurine (Tau) [130,131]L-glutamine (Gln) [132,133]L-arginine (Arg) [134,135,136]L-tryptophan (Trp) [137,138,139,140]

Immunometabolism, and specifically the impact of aminoacids on the immune response, capitalizes on the fact that T cells are highly environmentally influenced and that alterations in the nutritional state can modify their immunological response. Mechanistic research on NAFLD has mostly concentrated on the metabolism of carbohydrates and triglycerides; little is known, however, concerning the role of proteins and amino acids. Additionally, inconsistent findings were found in intervention and observational studies on the link between protein intake and NAFLD. Filing this knowledge gap could determine the ideal diet composition for NAFLD prevention and management.

In the increasingly pressing need to find solutions to the NAFLD burden, herbal medical treatments have also attracted considerable interest, and they are becoming increasingly popular in both Western and Eastern countries [141]. Traditional herb medicines (THM) have been widely employed for decades to treat NAFLD without actually fully understanding the underlying molecular mechanisms. With the modernization on THM, several clinical trials have been performed [142,143,144,145]. Among these, most of the research focuses on “*silymarin*”, an extract from milk thistle seeds, “*resveratrol*”, a non-flavonoid phenol produced by numerous plants, and “*curcumin*”, a pigment isolated from Curcuma longa Linn.

Silymarin, an extract from the dried seeds and fruits of the milk thistle plant (*Sylibun marianum*), has been shown to reduce oxidative stress and consequent cytotoxicity in preclinical studies, protecting intact liver cells as well as cells that have not yet been irreversibly damaged by oxidative stress [146].

Resveratrol (3,5,4′-trihydroxy-trans-stilbene) is a naturally derived phytoestrogen found in the skins of red grapes and berries such as grape peel and blueberries [147]. In their study, Poulsen and colleagues [143] showed how resveratrol increased fatty acid oxidation and reduced lipogenesis for reducing diet-induced hepatic fat accumulation. The underlying mechanism should still be fully elucidated, but, apparently, these effects are mediated by the activation of the AMPK/SIRT1 (AMP-activated protein kinase/sirtuin 1) axis [148].

Curcumin (Curcuma longa) is an active compound belonging to the curcuminoids family and is recognized as an antioxidant and anti-inflammatory agent [149]. In 2019, Saadati et al. investigated, in a double-blinded, randomized clinical trial (IRCT20100524004010N24), curcumin supplementation effects on inflammation and hepatic features in non-alcoholic fatty liver disease (NAFLD) [150]. Over a 12-week period, fifty NAFLD patients were randomly assigned to receive lifestyle modification advice, followed by 1500 mg of curcumin or a placebo. Curcumin supplementation was associated with a reduction in hepatic fibrosis (*p* 0.001) and nuclear factor-kappa B activity (NF-κB) (*p* 0.05) compared to the control group. Moreover HS, hepatic enzymes and tumor necrosis-α (TNF-α) were significantly reduced in both groups (*p*  <  0.05). Despite these results, no statistical changes have been detected between control and placebo groups.

Among the several measures examined, omega-3 fatty acids supplementation has proven highly efficient in liver steatosis clearing [151], specifically in the context of liver surgery [152]. Omega-3 long chain polyunsaturated fatty acids (*n*-3 LC-PUFA), mainly eicosapentaenoic acid (EPA, 20:5 *n*-3) and docosahexaenoic acid (DHA, 22:6 *n*-3), are accepted as being essential components of a healthy, balanced diet [152]. They are also considered to possess a sort of “inflammation-suppressive” effect and thus impact immunological reactions [153].

Omega-3 fatty acids (Ω3 FA) supplementation has been explored to ameliorate hepatic steatosis in both animal and human models in the setting of liver disease [154,155,156,157]. In particular, Clavien et al. demonstrated that Ω3 FA implementation can ameliorate ischemic/reperfusion injuries in ob/ob mice that undergo 45 min of segmental (70%) hepatic ischemia [154]. Via dynamic intravital fluoromicroscopy to analyze microvascular impairment, the authors showed a clear beneficial effect of Ω3 FA supplementation in ischemia/reperfusion injuries, with a decrease in microvascular dysfunction, biochemical hepatitis (AST) and inflammation. These results were later confirmed by Marsam et al. in 2013 in a rat model [155]. Three week methionine/choline-deficient diet rats with Ω3 FA supplementation underwent partial (70%) IR combined with the partial hepatectomy of the non-ischemic liver (30%), followed by 24 h of reperfusion. The control animal underwent the same surgical treatment, but with a standard diet without Ω3 FA supplementation. The treated group showed reduced ALT and TNF-α levels at different time-points associated with a global decrease in liver steatosis compared to the control group.

These preliminary data suggested that Ω3 FA has protective effects on I/R liver injuries during surgery, as well a potential use in clearing liver steatosis. Moreover, based on these promises, clinical investigations have been performed. In 2019, Linecker et al. published an elegant, multicenter, double-blind trial with a robust methodology (NTC01884948) [156]. A total of 261 patients undergoing liver surgery in three different centers (Zurich, Switzerland; Moscow, Russia and Bucharest, Romania) were randomly assigned to Omegaven^®^ supplementation (Ω3 FAs) or placebo (NaCl 0.9%) in a 1:1 allocation ratio the day before surgery. The results failed to demonstrate any advantages in morbidity and mortality protection derived from Ω3 FA supplementation.

The real beneficial effect of perioperative Ω3 FA supplementation within liver surgery settings is still controversial. In fact, a recent meta-analysis published by Xiao et al. analyzed five randomized control trials, concluding that Ω3 FA may just reduce the infection incidence of hepatic surgical procedures [157], without mentioning any advantage concerning liver steatosis clearing.

## 4. Illustrative Case

Herein, we report the favorable effect of a low-caloric, hyper-protein diet during a two-stage liver resection for colorectal metastases in a patient with severe steatosis.

We present a case of a 59-year-old Caucasian male (American Society of Anaesthesia score 2; Weight 94 kg; Body Mass Index 32.5 kg/m^2^; casual drinker) who was diagnosed with a sigmoid colon adenocarcinoma at 18 cm from the anal verge, accompanied by multiple synchronous and bilobar liver metastases. Physical examination revealed hepatomegaly (liver span measured at 24 cm). Liver disease work-up revealed the biochemical elevation of serum alanine aminotransaminase (ALT) (305 U/I) and aspartate aminotransferase (AST) (102 U/I). Any other cause (viral or autoimmune) possibly related to the increase in these values was ruled out. Neither diabetes nor hyperlipidemia run in the patient’s family history. After an excellent response to neo-adjuvant chemotherapy including 12 cycles of folinic acid/5-fluoruracil, oxaliplatin and irinotecan (FolfoxIri) followed by a steady line of 5-fluoruracil, a liver-first strategy with a two-stage liver resection was planned. Considering the criteria for steatosis on MRI (Figure 1a), a transjugular biopsy was performed and demonstrated severe macrovacuolar steatosis (up to 80%) and a hepatic venous pressure gradient of 6 mmHg. The first resection, a left parenchymal-sparing procedure, removed five metastases—all R0 with severe (90%) parenchymal macrovacuolar steatosis (Figure 1b). A right portal vein embolization was then completed (future remnant liver: 670 mL). Because of the severe steatosis, a hypocaloric (850 Kcal), hyper-protein diet (120 g of protein, 60 g of carbohydrates and 8 g of fat) (Scitec Nutrition^®^) was given 3 weeks prior to the second liver resection. The patient lost 7 kg (−7.44%; BMI 29.8 kg/m^2^). We mutuated this diet plan from a bariatric surgery program run by our center. This schedule consists of a high-protein drink in three out of the six meals, combined with a cup of milk (or yoghurt). Fiber supplements and a small portion of uncooked vegetables are provided for the remaining meals. A new MRI demonstrated a sharp decrease in the fat liver content (Figure 1c) that is estimated to be around 30%. Concurrently, basal liver tests also completely normalized (ALT 27 U/I; AST 25 U/I). A R0 right liver lobectomy was then performed. On histology, macrovacuolar steatosis had decreased to mild steatosis (30%) (Figure 1d). The patient recovered without complication and ultimately underwent a laparoscopic sigma resection three months later. During follow-up, a recurrence of colorectal liver metastases was found, and a new line of chemotherapy was then introduced (trifluridine and tipiracil), leading to stable disease. The patient remains alive 24 months after the sigma procedure.

## 5. Conclusions

Pre-operative liver optimization through nutritional therapy is increasingly important due to the epidemic of NAFLD and chemotherapy-induced liver injury. Severe steatosis represents one of the most significant risk factors for complications after major surgery. Ischemic/reperfusion injury and post-operative complications occur more often in livers with steatosis [119]. Prior to bariatric surgery, a hypo-caloric, hyper-protein diet has become standard practice for clearing steatosis and promoting liver shrinkage. Live liver donors can also be managed preoperatively with a calorie-controlled diet, exercise and/or drugs to improve hepatic parenchymal quality [120,121]. We herein document the benefits of a low-calorie diet in a two-stage liver resection process. Overall, the available literature, together with the present observation, support that the described 3-week hypocaloric hyper-protein diet can be used safely and efficiently in patients at risk of steatosis or with documented steatosis, especially before surgery. It could be indirectly linked to a decreased surgical risk profile.

Diet interventions are extremely heterogenous in terms of duration, delivery method and type [158,159].

Recently, Griffin et al. have reviewed the literature on dietary weight-loss strategies for adult patients prior to surgery. More specifically, they also have analyzed their impact on different surgical outcomes, including liver surgery [160]. From a total of 4553 records (derived from the MEDLINE, Embase, Cochrane and CINAHL databases), only 14 studies were included in the quality synthesis analysis. Among these, two trials focused solely on the benefits of a dietary regimen specifically prior to liver resection [91,117], as already discussed above. This highlights how, although this topic is already very timely and important, clinical data are still not adequate. Moreover, it should be kept in mind that the NAFLD trend is expected to rise rapidly in the near future, making hepatic pre-enhancement an increasingly hot topic in liver surgery.

## Figures and Tables

**Figure 1 nutrients-14-05340-f001:**
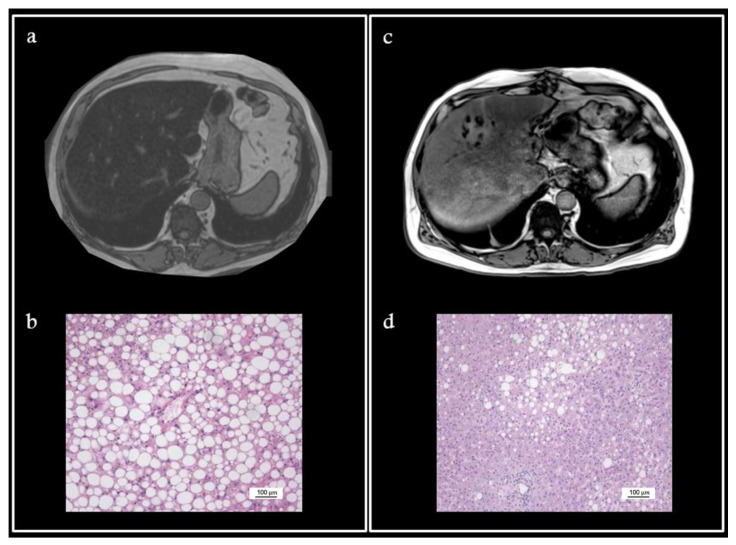
Preoperative MRI (T1-weighted) showed important steatosis (**a**). Liver biopsy confirmed 90% macrovacuolar steatosis ((**b**)-H&E). New MRI (T1-weighted) performed 3 weeks after the hypocaloric hyper-protein diet demonstrated a decrease in the fat liver content (**c**), with vacuolar steatosis decreased to 30% on histology ((**d**)-H&E).

**Table 1 nutrients-14-05340-t001:** Summary of the studies evaluating the impact of HS on liver surgery complications and outcomes.

Author	Year	Type of Study	Population Enrolled	Main Findings	Identifier	Ref.
Berhns et al.	1998	Retrospective	135 patients who had undergone major hepatic resection (4 or more liver segments)	HS has been associated to longer surgeries, higher rate of blood transfusion, post-operative bilirubine ans AST levels	PMID: 9841987	[86]
Pathak et al.	2010	Retrospective	102 patients undergoing hepatectomy for CRLM	HS does not influence post operative long-term survival	PMID: 19879103	[87]
Kooby et al.	2003	Retrospective matched case control	325 patients who had undergone hepatectomy for HCC, biliary cancer or CRLM	HS has been associated to higher rate of wound, hepatobiliary and gastro-intestinal complications. HS does not influence 5 yr survival rate.	PMID: 14675713	[88]
Fagenson et al.	2021	Retrospective propensity-score matched analysis	2927 patients undergoing major hepatectomy (3 or more liver segments)	HS has been associated with significant higher rate of biliary and pulmonary complications. HS has been conferred risk of postoperative mortality	PMID: 33358472	[89]
Nishio et al.	2015	Retrospective	518 HCC patients who underwent hepatic resection	Absence of HS has a significant impact on disease-free survival in non-b, non-c HCC patients	PMID: 25395147	[90]
Gomez et al.	2007	Retrospective	386 patients undergoing hepatic resection for CRLM	HS was associated with increased morbidity following hepatic resection	PMID: 17607707	[23]
Parkin et al.	2013	Retrospective	1793 patients who underwent first-time liver resection with background HS	HS was associated with improved 5 yr survival compared to normal background liver	UK charity chamber N°1054556	[22]
Reeves et al.	2013	Prospective	111 consecutive patients who had major elective hepatic resections	Short-term caloric restriction before liver resection significantly reduces both hepatic steatosis and steatohepatitis. Dietary modification also was associated with decreased intraoperative blood loss.	Darmouth Commitee for Protection of Human Subject n°22273	[91]
Barth et al	2019	Randomized	60 patients undergone liver liver surgey	Short-term, low-fat, and low-calorie diet significantly decreased blood loss with liver easier to manipulate	NCT01645852	[92]
Cauchy et al.	2013	Retrospective	560 patients undergoing liver resection for HCC	In presence of HS liver resection is still appropriate bau carries a high risk	PMID: 23147992	[93]
Koh et al.	2019	Retrospective analysis	996 patients who underwent liver resection for HCC	NAFLD-related HCC is associated with greater surgical morbidity and post-hepatectomy liver failure. Despite this, long-term survival outcomes are favorable compared with non-NAFLD etiologies	PMID: 31398386	[7]

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
