# Peer review of "Clearing Steatosis Prior to Liver Surgery for Colorectal Metastasis: A Narrative Review and Case Illustration"

_nutrients, 2022, doi:10.3390/nu14245340_

Round 1
Reviewer 1 Report
The manuscript submitted to Nutrients by Peloso et al., titled: "Clearing steatosis prior to liver surgery for colorectal metastasis: A narrative review and case illustration", is an interesting review and carries significant value from a clinical perspective. The manuscript is well written and very well organized and presented. It flows well and makes it easy for the reader to follow.
One potentially interesting concept to be discussed in the discussion section briefly is the use of diet with a specific focus on amino acids in terms of cancer control. An interesting paper towards that end is below.
Sikalidis AK (2015) Amino Acids and Immune Response: A role for cysteine, glutamine, phenylalanine, tryptophan and arginine in T-cell function and cancer? Pathol Oncol Res. 21(1):9-17. doi: 10.1007/s12253-014-9860-0.
Good job overall.
Author Response
First of all, we would like to thank the Editorial team and the reviewers for the valuable suggestions to strengthen our work, and for the opportunity to revise our paper. Here is a point-to-point response to the reviewers’ comments and concerns. All the amendments can be seen in the revised version of the article highlighted in yellow.
We do hope that the new version of the article will meet everybody’s expectations.
With our very best regards,
The Authors
----------------------------------------------------------------------------------------
R: One potentially interesting concept to be discussed in the discussion section briefly is the use of diet with a specific focus on amino acids in terms of cancer control. An interesting paper towards that end is below.
Sikalidis AK (2015) Amino Acids and Immune Response: A role for cysteine, glutamine, phenylalanine, tryptophan and arginine in T-cell function and cancer? Pathol Oncol Res. 21(1):9-17. doi: 10.1007/s12253-014-9860-0.
A: Thanks for your comment and input.Immunometabolism is gaining crucial attention in the last decade and we agree that adding a paragraph on this subject would increase the quality of our manuscript.
Following your suggestion we have added this at the end of the paragraph "clearing steatosis prior liver surgery" highlighting the most important aminoacids in immunometabolism as well their "potential impact" on NAFLD management
For Rev1 changes are highlighted in yellow
Reviewer 2 Report
Please find my comments as mentioned below:
1) The manuscript consists of minor grammatical and spelling errors like on page 9 line 372. Kindly recheck the whole manuscript and modify the errors. Both US and British English are used. Kindly maintain a uniform language pattern.
2) A list of abbreviations should be included.
3) Mention some articles stating the correlation between NAS score and colorectal metastasis.
4) Page 9 line 373 describes that it takes 4-12 months for medications to manage steatosis. What about the use of herbal drugs along with lifestyle modifications?
5) Mention some articles wherein dietary modifications along with herbal medicines have been used to reduce fatty liver.
6) The reference format is not the same. Some contain doi, some PMID, and some devoid of both. Maintain a uniform format.
7) Please mention the NCT number in Table 1 for the respective trials.
Author Response
We thank the Editorial team and the reviewer for the valuable suggestions to strengthen our work, and for the opportunity to revise our paper. We do hope that the new version of the article will meet everybody’s expectations.
With our very best regards,
Here is a point-to-point response to the reviewers’ comments and concerns (green highlighted)
---------------------------------------------------------------------------------------------------------------------------
Please find my comments as mentioned below:
R: The manuscript consists of minor grammatical and spelling errors like on page 9 line 372. Kindlyrecheck the whole manuscript and modify the errors. Both US and British English are used. Kindlymaintain a uniform language pattern.
A: We would like to thank the reviewer for the observations. The whole manuscript has been now revised by an English mother-tongue scientific writer
R: A list of abbreviations should be included.
A: A list of abbreviation has been added at the end of the title page
R: Mention some articles stating the correlation between NAS score and colorectal metastasis.
A: Thanks for this input. We have now added several articles concerning NAS score and CRLM in the paragraph titled “Influence of metabolic disease and NAFLD on colorectal cancer”
R: Page 9 line 373 describes that it takes 4-12 months for medications to manage steatosis. What about the use of herbal drugs along with lifestyle modifications? Mention some articles wherein dietary modifications along with herbal medicines have been used to reduce fatty liver.
A: We thank the reviewer for raising this point. Herbal medical treatment is gaining attention for NAFLD treatment and, accordingly to reviewers’ suggestion, we added different papers concerning this field (from Ref 138 to 147)
R: The reference format is not the same. Some contain doi, some PMID, and some devoid of both. Maintain a uniform format.
A: The whole Refences body has been revised. According to the reviewer's recommendation, all the references are now uniformly formatted (NLM writing style)
R: Please mention the NCT number in Table 1 for the respective trials.
A: Thanks for the input. Your suggestion led us to revisit the Table 1, which has been entirely re-edited and fine-tuned. A new 'identifier' column has been added and, where possible, the NCT code entered. Otherwise, to allow the reader a quick interpretation, we have used the PMID code
Reviewer 3 Report
The authors propose an interesting topic.
However, there are many conditions to review in order to consider it:
- The whole manuscript is addressed to a path before a tumor-related surgery, so everything related to the presentation of the case should be related to the tumor and not in general to NAFLD; however, the presentation part should be reduced.
- Is the appearance of the microbiota very hypothetical and above all is it the condition of NAFLD that alters the microbiota or vice versa?
- The clearing process, which would be very interesting, is reduced to a single paragraph, it should be rewritten and expanded a lot since it is the focus of the manuscript.
- The diet presented in ref 114 is a ketogenic one, so it should be presented and discussed
- The use of supplements such as omega3 should also be better discussed
- In the case report a still different program was used, which is high-protein but with a non-negligible amount of carbohydrates and which probably did not trigger ketosis, why was this choice made? Was ketonemia measured? The case must be presented better, nothing is said about the biochemical-clinical parameters, nor about the body composition, certainly, the biopsy shows an improvement, but it is not enough
- The conclusions should be rewritten, considering the weaknesses and merits of the possible intervention, but correlating it better to what the state of the art is and better describing what has been done.
Author Response
The authors propose an interesting topic.
However, there are many conditions to review in order to consider it:
R: The whole manuscript is addressed to a path before a tumor-related surgery, so everything related to the presentation of the case should be related to the tumor and not in general to NAFLD; however, the presentation part should be reduced.
A: We thank the reviewer for raising this point. The primary aim of our manuscript is to explore ways to handle the underlying fatty liver disease in order to improve quality of the management of patients with NAFLD, other than focusing on tumor-related surgery. We revised the text accordingly.
R: Is the appearance of the microbiota very hypothetical and above all is it the condition of NAFLD that alters the microbiota or vice versa?
A: We thank the reviewer for his/her valuable concern and for raising this critical issue. In our manuscript we just try to illustrate how microbiota should be identified as a potential player for NAFLD. At the same time bacterial dysbiosis is undoubtedly also a result of NAFLD presence, in a sort of vicious circle. We have revised the text accordingly.
R: The clearing process, which would be very interesting, is reduced to a single paragraph, it should be rewritten and expanded a lot since it is the focus of the manuscript.
A: Thank you so much for this suggestion. The whole paragraph has been implemented and expanded, also following suggestions deriving from reviewers no°1 and no°2. Moreover as suggested we also added most important references about the Omega3 supplementation.
R: The diet presented in ref 114 is a ketogenic one, so it should be presented and discussed
A: We thank the reviewer for his/her suggestion which allowed us to better detail the diet regimes explored in Ref.114 and Ref.115 and furthermore to correct some typos. Table no°1 has been revised and implemented accordingly
R: The use of supplements such as omega3 should also be better discussed
A: We thank the review for the input. Omega3 supplementation have been proposed for the prevention/treatment of several diseases including hepatic diseases and liver surgery. Accordingly, we added different papers concerning this field (from Ref 149 to 155) both on animal and clinical models.
R: In the case report a still different program was used, which is high-protein but with a non-negligible amount of carbohydrates and which probably did not trigger ketosis, why was this choice made? Was ketonemia measured? The case must be presented better, nothing is said about the biochemical-clinical parameters, nor about the body composition, certainly, the biopsy shows an improvement, but it is not enough
A: We thank the reviewer for the valuable points. Concerning the diet regime used, we have derived this diet plan from than one used in our bariatric surgery program, which allowed a significant liver shrinkage before bariatric surgery. Unfortunately, we didn’t measure ketonemia. The description of the case report has been (we believe) significantly improved. More details concerning the diet plan have been added as well biochemical markers of liver function.
R: The conclusions should be rewritten, considering the weaknesses and merits of the possible intervention, but correlating it better to what the state of the art is and better describing what has been done.
A: We really thank the reviewer for the suggestion. Conclusion section has been rewritten and implemented also considering the most recent literature (Ref. 160).
In the text, modifications for Rev no°3 are blue out-lighted.

Round 2
Reviewer 3 Report
I think that the authors made enough improvements to the manuscript.